# Spatial heterogeneity of coral reef benthic communities in Kenya

**Juliet Furaha Karisa** [1,2,3,4], **David Obare Obura** [5], **Chaolun Allen Chen** [1,2,3,6,7]*

**1** Biodiversity Research Center, Academia Sinica, Taipei, Taiwan, **2** Biodiversity Program, Taiwan International Graduate Program, Academia Sinica and National Taiwan Normal University, Taipei, Taiwan, **3** Department of Life Science, National Taiwan Normal University, Taipei, Taiwan, **4** Kenya Marine and Fisheries Research Institute, Mombasa, Kenya, **5** CORDIO East Africa, Mombasa, Kenya, **6** Institute of Oceanography, National Taiwan University, Taipei, Taiwan, **7** Department of Life Science, Tung Hai University, Taichung, Taiwan

* cac@gate.sinica.edu.tw

**Data Availability Statement:** All relevant data are within the manuscript and its Supporting Information files.

**Funding:** This work benefited from various funding sources: JK was a recipient of a scholarship funded

## Abstract

Spatial patterns of coral reef benthic communities vary across a range of broad-scale bio-geographical levels to fine-scale local habitat conditions. This study described spatial patterns of coral reef benthic communities spanning across the 536-km coast of Kenya. Thirty-eight reef sites representing different geographical zones within an array of habitats and management levels were assessed by benthic cover, coral genera and coral colony size classes. Three geographical zones were identified along the latitudinal gradient based on their benthic community composition. Hard coral dominated the three zones with highest cover in the south and *Porites* being the most abundant genus. Almost all 15 benthic variables differed significantly between geographical zones. The interaction of habitat factors and management levels created a localised pattern within each zone. Four habitats were identified based on their similarity in benthic community composition; 1. Deep-Exposed Patch reef in Reserve areas (DEPR), 2. Deep-Exposed Fringing reefs in Unprotected areas (DEFU), 3. Shallow Fringing and Lagoon reefs in Protected and Reserve areas (SFLPR) and 4. Shallow Patch and Channel reefs (SPC). DEPR was found in the north zone only and its benthic community was predominantly crustose coralline algae. DEFU was found in central and south zones mainly dominated by soft corals, *Acropora*, *Montipora*, juvenile corals and small colonies of adult corals. SFLPR was dominated by macroalgae and turf algae and was found in north and central zones. SPC was found across all geographical zones with a benthic community dominated by hard corals of mostly large colonies of *Porites* and *Echinopora*. The north zone exhibits habitat types that support resistance properties, the south supports recovery processes and central zone acts as an ecological corridor between zones. Identifying habitats with different roles in reef resilience is useful information for marine spatial planning and supports the process of designing effective marine protected areas.

by the Taiwan International Graduate Program–
Biodiversity Program of Academia Sinica and
National Taiwan Normal University, Taiwan. CAC
was supported by funding from Biodiversity
Research Centre, Academia Sinica and Ministry of
Science and Technology, Taiwan. Fieldwork was
partly funded by the Western Indian Ocean Marine
Science Association (WIOMSA) through the Marine
Research Grant (MARG) program under grant
number MARG I Contract 006/2014 and The
Nature Conservancy Africa Regional Office.
Logistical support for fieldwork was provided by
the Kenya Marine and Fisheries Research Institute
(KMFRI), Kenya Wildlife Service (KWS) and World
Wide Fund for Nature Kenya (WWF-Kenya). The
funders had no role in study design, data collection
and analysis, decision to publish, or preparation of
the manuscript.

**Competing interests:** The authors have declared
that no competing interests exist.

## Introduction

Coral reefs are among the most biodiverse ecosystems in the world [1], providing an array of
ecological services which are important for human well-being [2, 3]. However, coral reefs are
declining at an alarming rate globally with climate change posing as one of the biggest threats
[4]. There are still limited global efforts to abate climate change [5] making reefs a highly
threatened ecosystem with climate scenarios showing 99% of the reef could disappear in this
century [6]. The persistence of coral reefs will rely heavily on their ability to maintain a coral-
dominated state and avoid shift to algal-dominated or other alternative stable states amid the
inevitable effects of climate change [7–11].

The Kenya coast stretches along 536 km, between latitudes $1^0$ and $5^0$ S with narrow fringing
reefs in the southern part and patchy reefs with low reef development in the north [12, 13],
(Fig 1). The distribution of coral species in Kenyan reefs is influenced by the large-scale cur-
rent dynamics with the East African Coastal Current (EACC) bringing coral larvae from the
southern 'center of diversity' for the Western Indian Ocean (WIO) region [14]. A cold-water
system prevails in northern Kenya due to the convergence of the EACC with the seasonal
Somali Current (SC) that is characterized by poor water conditions for reef development [15,
12]. The interaction of the EACC and the SC in the north creates a marginal, high-latitude and
upwelling system with transitioning communities from the East African to Somali-Arabian
fauna [16]. This results in high coral species diversity in the southern parts of Kenya and a
decrease in diversity towards the north [16]. In addition, the presence of river systems in the
central-northern region introduces small-scale influences in species distribution by creating
environmental barriers that further limit the transport of larvae to the north [17].

There has been a substantial amount of coral reef studies in Kenya focusing on a range of
issues including: the effect of MPAs in protecting habitats and biodiversity [18–20], impact of
anthropogenic stressors, including the 1998/99 mass coral bleaching event, on coral and fish
communities [21–25]; but without reference to spatial patterns of communities along the lati-
tudinal gradient. These studies, which include coral reef monitoring with different levels of
management programs (marine reserve vs. marine park), have been running for more than
three decades showing long-term trends in coral reef communities [26, 2, 27] but have mostly
focused on shallow lagoons within the south coast, leaving deeper forereefs and the north zone
depauperate of studies. In addition, there is limited information on coral reef communities on
reefs that are remote and not within conservation areas along the Kenyan coast. Knowing the
spatial patterns of the coral reef benthic community can help identify habitats that offer resis-
tance/protection to, and recovery after a disturbance including climate change and could help
the persistence of coral reefs.

In this study, we follow a quantitative approach to characterize coral reef benthic communi-
ties along the Kenyan coast. Using extensive data collected at different geographical zones, reef
habitats and management levels multivariate analysis was utilized to investigate the effects of
the latitudinal gradient, local habitat and management in structuring spatial patterns in ben-
thic communities.

## Materials and methods

### Study area

Kenya is located on the east coast of Africa between Somalia and Tanzania with a coastline
running approximately 500km long between $1^0$ and $5^0$ S (Fig 1), [12]. Two monsoons namely,
southwest (December to March) and northeast (May to October) highly influence the climate
and seasonality of this area. The Kenyan coast also experiences the EACC originating from

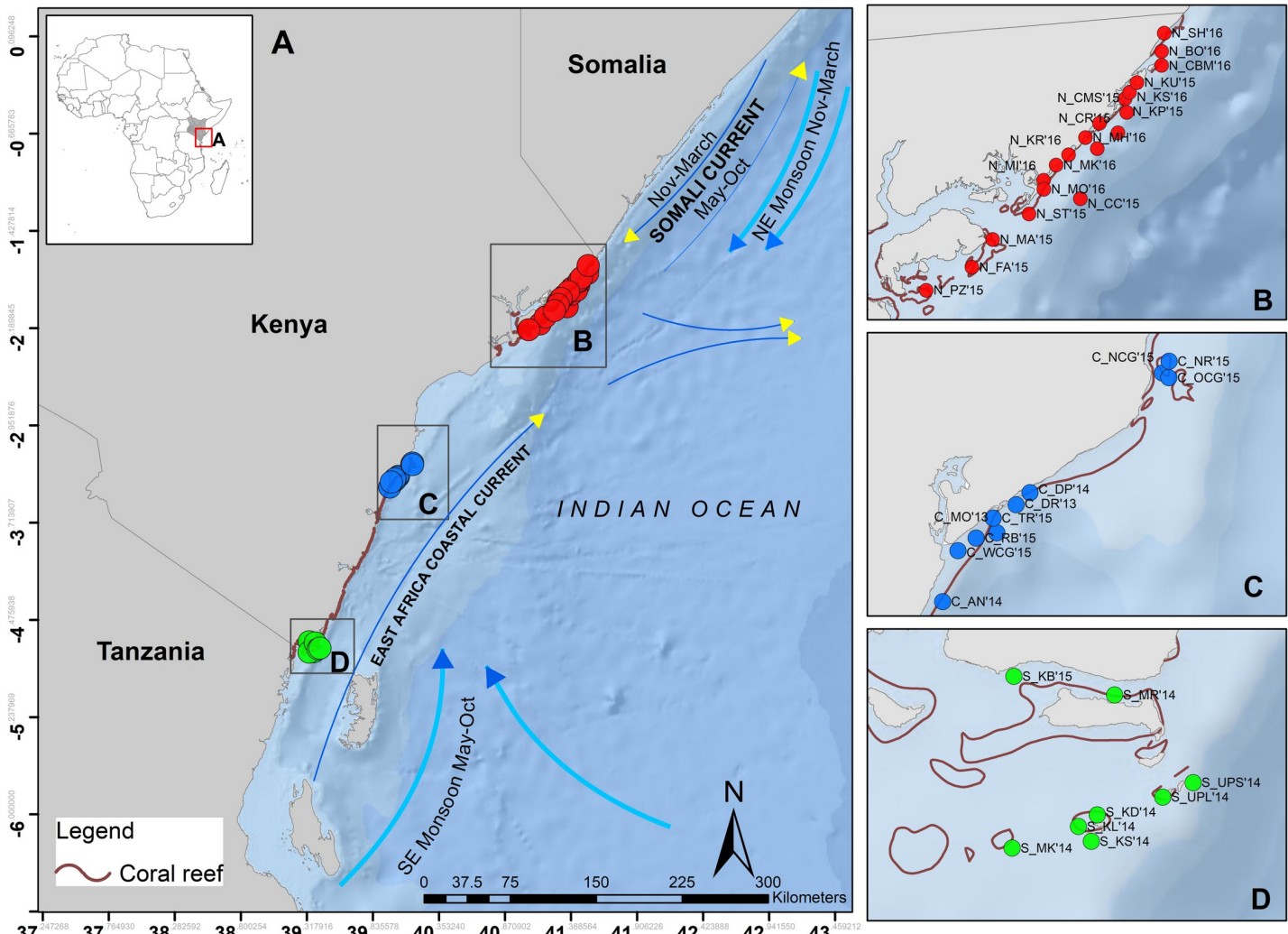

**Fig 1. Study area.** (A) Locations of the study sites along the Kenyan coast divided into three geographical zones in the, (B) North (red), (C) Central (blue) and (D) South (green). Ocean base map credits- Esri, Garmin, GEBCO,NGDC, and other contributors.

Tanzania and flows northwards throughout the year. In addition, the SC mainly driven by northeast monsoon (NEM) counteracts the EACC when the wind blows southwards. Interaction of these two currents during the NEM causes cooling in the northern part of Kenya, and through mixing of nutrients makes this zone highly productive [28].

Coral reefs in Kenya are characterized by narrow fringing reefs in the southern part and patchy and discontinuous fringing reefs in the northern part [12]. These reefs are found at different depths ranging from less than 1 m at low tide to over 20 m. There are different geomorphological characteristics of the reefs with channel reefs in highly sedimented areas of river inflows to ocean, lagoon reefs mostly occurring at leeward side of reef flat and fringing reefs on the seaward side and patch reefs occurring at offshore outcrops of reefs.

Sea surface temperature (SST) for two 'normal years' with no influence from extreme climate events (e.g.the Indian Ocean Dipole) in 2013 and 2014 show that the north zone had a low mean SST (27.45 $^{0}$C) but with the highest variation with a standard deviation of 1.23 $^{0}$C, a maximum of 30.16 $^{0}$C and minimum of 25.27 $^{0}$C. The south zone had the highest mean SST of

27.63 $^{0}$C and the lowest variation (sd = 1.15 $^{0}$C, max = 29.71 $^{0}$C, min = 25.71 $^{0}$C). The central zone had the lowest mean SST (27.26 $^{0}$C) and an intermediate variation (sd = 1.21 $^{0}$C, max = 29.58 $^{0}$C, min = 25.41 $^{0}$C). SST data were derived from MODIS satellite source by plotting georeferenced surveyed sites on a GIS system. Using ArcGis 10.8, points were created along vector lines at two pixels apart (one pixel = 4x4km) for each zone (north = 10points, central = 5points and south = 3points) and SST values were averaged over the two years. There are different levels of management for coral reefs in Kenya; 'marine parks' receive legal protection from fishing activities and collection of shells or corals. 'Marine reserves' receive partial protection with only traditional fishing methods allowed. 'Unprotected' areas are only regulated by fishing licensing and recently some areas receive fishing restrictions set by local communities. The southern zone has 2 parks and 2 reserves, and the central zone 1 park and 1 reserve in south zone. The rest of the reef is within unprotected areas. The northern part has only 1 reserve and the rest are unprotected reef areas.

## Sampling design

A hierarchical sampling design (S1 Table) was used to collect benthic community data across Kenya coast. There were three pre-defined geographical zones (North, Central, South). In each geographical zone, sampling sites were selected to reflect different habitat factors: two depth levels (deep [6.5-18m], shallow [0-6m]), two levels of exposure to oceanic waves (exposed, sheltered) and four different reef types (channel, lagoon, fringing, patch). Additionally, three different management states were sampled; park (no-extraction), reserve (regulated extraction) and unprotected (extraction allowed). It should be noted that not all habitat types and management levels were found at the three geographical zones, for example the north zone had sites in reserve and unprotected levels but there is no park in this zone (S1 Table). Furthermore, the north zone had all the four levels of reef type while central and south zones did not. In the end, there were twenty sites in the north, ten sites in central and eight sites in the south. Choice of these study sites was based on historical monitoring of reef resilience [29], with an additional fourteen sites selected haphazardly and stratified within each geographical zone in order to cover habitats that had not been studied before.

## Data collection

Surveys for benthic communities were conducted during the north-east monsoon season during 2015 and 2016. A modified resilience assessment protocol [29] was used to record benthic cover and coral size class distributions. Benthic cover was recorded using photoquadrats with an underwater camera positioned at approximately 1 m above the seafloor with the aid of a 1 m PVC stick. Within each belt-transect, twenty five 50 x 50 cm photos were taken at an interval of 1 m. Cover was recorded in fifteen categories; hard corals (HC), coralline algae (CA), macroalgae (MA), turf algae (TA), soft corals (SC), *Halimeda* (HA), bare substrate (BS), dead standing coral (DSC), recently dead coral (RDC), invertebrates (INV), rubble (RB), sand (SD), seagrass (SG), silt (SL) and others (OT). HA was separated from macroalgae group as it is a calcifying algae that contributes to the production of carbonate and formation of reef sediments [30, 31]. Living coral cover (LCC) was recorded by coral genera.

The size class distribution of coral colonies was measured for 23 selected genera of adult corals using the belt transects [29], and for all juvenile corals by subsampling within the belt transects in six 1m$^2$ quadrats that were 5 m apart along the 25m belt. Adult colonies were defined as those larger than 10cm in diameter and grouped into six size classes (11–20 cm, 21–40 cm, 41–80 cm, 81–160 cm, 161–320 cm and >320 cm). Juvenile corals were defined as

those smaller than 10 cm in diameter and were classified into three size classes (1–2.5 cm, 2.6–5 cm and 6–10 cm).

Benthic photos were analysed using CPCe v4.0 software [32] by randomly placing 25 points on each photo. This number of points overlaid on the photo are within the optimum number required for accurately estimating coral cover in a coral reef that has about 30% cover [33]. The type of benthic substrate was identified under each point.

## Data analysis

Three datasets were analyzed: 1. benthic cover by major categories (S1 Table), 2. benthic cover by coral genera (S2 Table) and 3. coral size classes (S3 Table). A set of criteria was established to select only the key variables important for determining the spatial pattern as follows. First, only biotic variables from the major benthic categories were considered as these are influenced by latitudinal gradient and habitat environmental conditions. Only coral genera that had a > 1% cover were included in the analysis. In the second stage, analysis was done in PRIMER v7 [34] where variables were square root transformed in order to reduce skewness and make the variances more homogenous. Each of the three datasets were examined by principle coordinate analysis (PCoA) in order to select those key variables that only contributed to the clustering of sites. Only those variables whose vector direction demonstrated influencing on clustering of sites were selected for further analyses. Variables whose vectors had similar direction were either pooled together or representative variables were selected to avoid analysing many variables that otherwise have similar effect on clustering sites. A permutational multivariate analysis of variance (PERMANOVA) [35], was conducted for each of the three datasets to determine if geographic zone had an effect on site clustering. The selected key variables from the three datasets were then pooled together and collectively analysed. A One-way ANOVA was done in excel to determine how each key variable differed across the geographic zones. PERMANOVA was done in R using the Adonis function within the vegan package [36] to test the effect of depth, reef type, exposure to oceanic waves, management and their interaction. Since geographic zone was found to have a significant effect on variables, this analysis was treated as a block design where all other factors were randomly assigned within each block (i.e. geographic zone).

## Results

### Description of variables-overall

Among the major benthic categories assessed, there were ten biotic benthic communities with hard corals having the highest cover (24.1 ± 26.7%) followed by turf algae (16.2 ± 21.8%) and macroalgae (12.9 ± 22.7%), (S2 Table). Forty-three coral genera were recorded with only 13 having a cover of > 1%. *Porites* was the most abundant genus (7.7 ± 17.6%) followed by *Acropora* (2.1 ± 7.3), *Echinopora* (2.06 ± 10.07) and *Montipora* (1.64 ± 7.99) (S3 Table). Coral density (number of colonies per 100 m$^2$) was higher among juvenile coral colonies and some of the adult colonies within small size classes. Density was highest in the 6–10 cm size class (197.1 ± 182.1 colonies) and lowest in the >320 cm size class (5.7 ± 10.4 colonies) (S4 Table).

### Selection of key variables

Three PCoAs performed on the ten biotic benthic categories, 13 coral genera and ten coral size classes showed clustering of sites (Fig 2A–2C). Of the ten benthic categories, six (hard corals, coralline algae, turf algae, macroalgae, soft corals and dead standing corals), were selected for further analyses (Fig 2A). *Halimeda* was not included because their vectors pointed in the

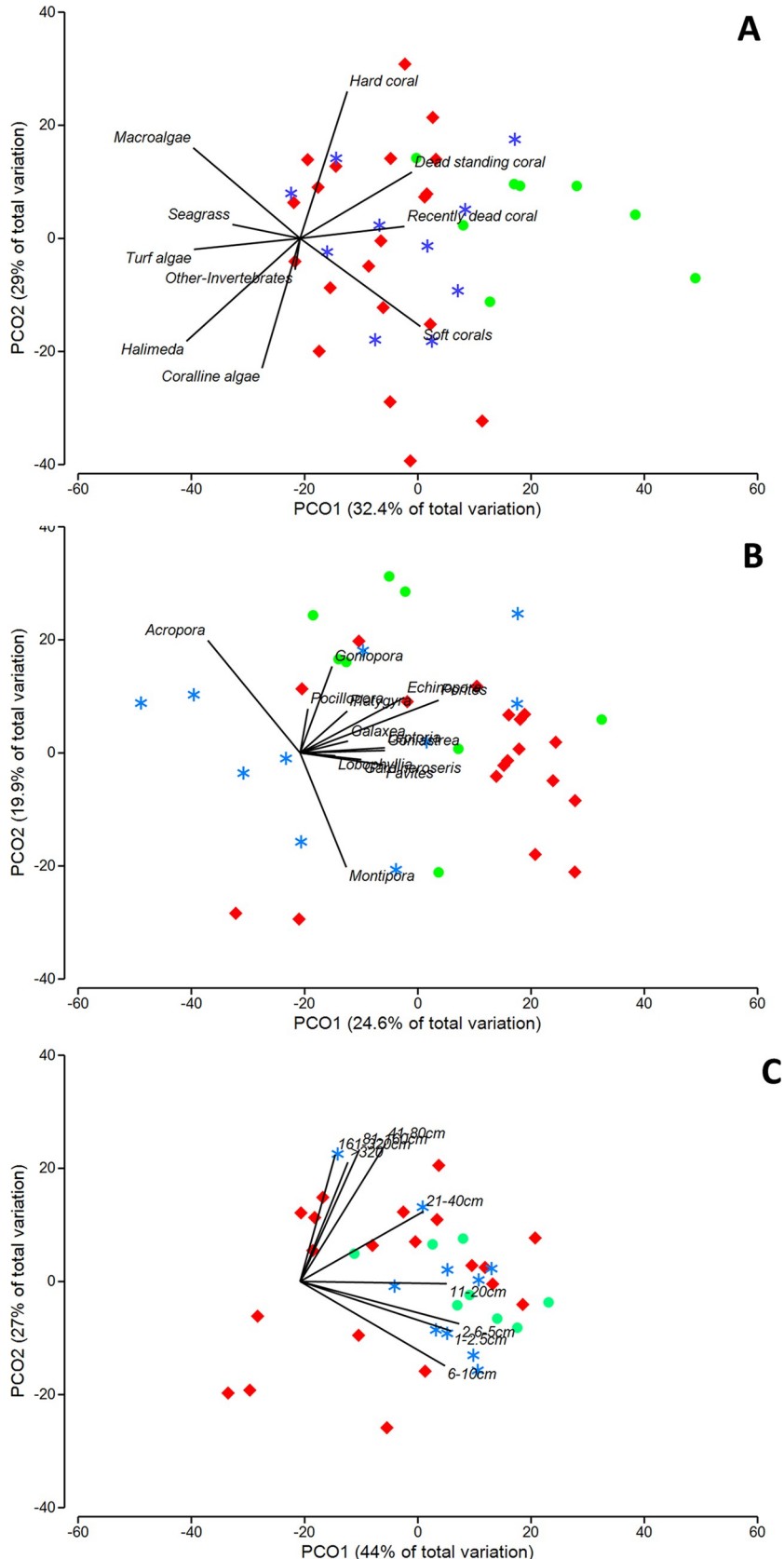

**Fig 2.** Principle Coordinate analysis (PCoA) analyses of (A) major benthic communities (B) coral genera and (C) coral size classes. Clusters of 38 reef sites based on Bray–Curtis index. Different colours differentiate sites based on the geographical zone (red = north, blue = central, green = south).

same direction, so only CCA was selected as the representative variable. Seagrass and 'other-invertebrate' categories were not included as key variables because they had very low mean percentage cover. Only four representative genera, *Acropora*, *Montipora*, *Echinopora and Porites*, were considered as key variables for further analysis based on the different sets of similar-effect vectors on the PCoA (Fig 2B) as well as their abundance (S6 Table). Vectors of *Acropora* and *Montipora* clearly pointed in opposite direction showing that they had distinct influence on site clustering. Vectors of *Porites* and *Echinopora* pointed in similar direction and most orthogonal to *Acropora* and *Montipora*, and were among the most abundant genera recorded in this study, so were selected as indicative of other genera. Vectors for 1–10 cm size classes were closely grouped, as well as vectors for > 40 cm size classes (Fig 2C), so a representative size class from each was selected. The other size-classes (11-20cm and 20-40cm) were considered separately based on their independence in clustering sites.

A PERMANOVA test on the effect of geography for each of the datasets showed significant differences between zones (S8 Table) and therefore further analysis was restricted to within geographic zones (Table 1).

## Describing key variables

One-way ANOVA for the key variables (Fig 3) showed that almost all differed significantly ($p<0.05$) across geographical zones. There was significant difference in hard coral cover across the three geographical zones ($F_{2,773} = 6.88$, $p = 0.00$, Table 2) with the highest cover occurring in the south ($32.06 \pm 29.48$) while north and central zones had similar cover ($22.5 \pm 26.6\%$ and $23.0 \pm 23.7\%$, respectively), (Fig 3). Crustose coralline algae cover was significantly higher in the north ($8.9 \pm 14.8\%$) than in the central ($3.4 \pm 8.0\%$) and south ($2.3 \pm 5.6\%$)($F_{2,773} = 21.02$, $p = 0.00$). Turf algae had a significantly higher cover in north and central zones ($17.8 \pm 22.7\%$ and $19.2 \pm 22.5\%$, respectively) and lowest in the south ($6.3 \pm 12.8\%$)($F_{2,773} = 16.85$, $p = 0.00$). Soft coral cover was significantly higher in the south ($17.9 \pm 23.1\%$) followed by central ($8.4 \pm 16.0\%$) and lowest in the north ($3.7 \pm 12.2\%$)($F_{2,773} = 16.85$, $p = 0.00$). Dead standing coral was significantly higher ($F_{2,773} = 15.19$, $p = 0.00$) in the south ($1.63 \pm 4.5\%$) followed by central ($0.7 \pm 3.0\%$) and north ($0.2 \pm 1.6\%$). Macroalgae cover did not show significant variation across geographic zones.

There was significant difference in the cover of *Porites* across the three geographic zones ($F_{2,774} = 3.6$, $p = 0.03$, Table 2) with highest cover in the south ($10.3 \pm 17.9\%$) followed by north ($7.9 \pm 18.2\%$) and central ($4.7 \pm 15.0\%$) (Fig 3). *Acropora* cover was significantly different across zones ($F_{2,774} = 24.8$, $p = 0.00$) with highest cover in south ($5.9 \pm 11.2\%$) followed by central ($2.6 \pm 6.6\%$) and north zones ($1.0 \pm 5.7\%$). *Montipora* cover differed significantly across the zones ($F_{2,774} = 5.13$, $p = 0.01$) with the highest cover in the south ($3.3 \pm 11.5\%$) followed by north ($1.6 \pm 8.0\%$) and central zones ($0.4 \pm 2.2\%$). *Echinopora* cover did not show any difference across geographic zones.

A total of 64,850 coral colonies were recorded with an average density of 79 colonies per 100m$^2$ across 4 size classes. The density of corals (number of colonies per 100m$^2$) in 1-10cm size class differed significantly ($F_{2,88} = 5.0$, $p = 0.01$; Table 2) with the highest density in central and south zones ($510.8 \pm 333.0$ colonies and $510.9 \pm 352.8$ colonies, respectively) (Fig 3). The density of 11-20cm size class was significantly higher in the south zone ($208.3 \pm 134.5$ colonies) followed by central and north zones ($108.6 \pm 55.3$ and $107.5 \pm 92.8$ colonies,

**Table 1. Summary of study sites at three geographical zones.** Site habitat characteristics (depth, exposure, reef type) and management level including their corresponding number of transects surveyed for each benthic community along the Kenyan coast.

| Geographic zone | Site name | Site code | Latitude | Longitude | Depth | Exposure | Reef type | Management | No. of transects | | |
| | | | | | | | | | Benthic cover | Coral genera | Coral size-class |
|---|---|---|---|---|---|---|---|---|---|---|---|
| North | Chole | N_CH'15 | 41.3831 | -1.8929 | Shallow | Exposed | Lagoon | Reserve | 25 | 25 | 2 |
| | Fawacho | N_FA'15 | 41.1465 | -2.1567 | Shallow | Sheltered | Channel | Unprotected | 24 | 24 | 3 |
| | Kupi | N_KP'15 | 41.4379 | -1.8329 | Shallow | Sheltered | Lagoon | Reserve | 24 | 24 | 2 |
| | Mabiyu | N_MA'15 | 41.1871 | -2.1024 | Shallow | Exposed | Channel | Unprotected | 24 | 24 | 2 |
| | Mikes Inner | N_MI'16 | 41.2853 | -1.9890 | Shallow | Sheltered | Channel | Reserve | 25 | 25 | 2 |
| | Mikes Outer | N_MO'16 | 41.2920 | -1.9945 | Shallow | Exposed | Patch | Reserve | 25 | 25 | 2 |
| | Mkokoni | N_MK'16 | 41.3044 | -1.9636 | Shallow | Sheltered | Lagoon | Reserve | 25 | 25 | 2 |
| | Mlango wa Muhindi | N_MH'16 | 41.3714 | -1.9079 | Shallow | Exposed | Fringing | Reserve | 25 | 25 | 2 |
| | Shimo La Tewa | N_ST'15 | 41.2467 | -2.0430 | Shallow | Exposed | Fringing | Reserve | 24 | 24 | 2 |
| | Boso | N_BO'16 | 41.5181 | -1.7337 | Shallow | Sheltered | Lagoon | Reserve | 25 | 25 | 2 |
| | Kishanga | N_KS'16 | 41.4366 | -1.8283 | Shallow | Sheltered | Lagoon | Reserve | 25 | 25 | 2 |
| | Kui | N_KU'15 | 41.4386 | -1.8224 | Shallow | Sheltered | Lagoon | Reserve | 24 | 24 | 2 |
| | Chongo cha Bomani | N_CBM'16 | 41.5181 | -1.7611 | Deep | Exposed | Patch | Reserve | 25 | 25 | 2 |
| | Kwa Radi | N_KR'16 | 41.3534 | -1.9192 | Shallow | Exposed | Fringing | Reserve | 25 | 25 | 2 |
| | Pezzali | N_PZ'15 | 41.0569 | -2.2018 | Shallow | Exposed | Fringing | Unprotected | 24 | 24 | 2 |
| | Shili | N_SH'16 | 41.5232 | -1.6980 | Shallow | Sheltered | Fringing | Reserve | 25 | 25 | 2 |
| | Chongo cha Chano | N_CC'15 | 41.3330 | -2.0107 | Deep | Exposed | Patch | Reserve | 25 | 25 | 2 |
| | Chongo cha Mvundeni | N_CM'15 | 41.3913 | -1.9343 | Deep | Exposed | Patch | Reserve | 25 | 25 | 2 |
| | Chongo cha Mwongo Shariff | N_CMS'15 | 41.4496 | -1.8678 | Deep | Exposed | Patch | Reserve | 23 | 23 | 2 |
| | Chongo cha Rubu | N_CR'15 | 41.4179 | -1.9034 | Deep | Exposed | Patch | Reserve | 25 | 25 | 2 |
| Central | New Coral Gardens | C_NCG'15 | 40.1411 | -3.2559 | Shallow | Sheltered | Patch | Park | 24 | 24 | 2 |
| | Anthias | C_AN'14 | 40.0316 | -3.6209 | Deep | Exposed | Fringing | Unprotected | 12 | 12 | 3 |
| | Dolphin | C_DP'14 | 40.0241 | -3.3670 | Deep | Exposed | Fringing | Unprotected | 12 | 12 | 3 |
| | Drummers | C_DR'13 | 40.0180 | -3.3695 | Deep | Exposed | Fringing | Unprotected | 12 | 12 | 3 |
| | Moray | C_MO'13 | 40.0107 | -3.3788 | Deep | Exposed | Fringing | Unprotected | 12 | 12 | 3 |
| | North Reef | C_NR'15 | 40.1464 | -3.2467 | Shallow | Exposed | Patch | Park | 24 | 24 | 2 |
| | Old Coral Gardens | C_OCG'15 | 40.1461 | -3.2596 | Shallow | Sheltered | Patch | Park | 24 | 24 | 2 |
| | Richard Bennette | C_RB'15 | 39.9964 | -3.3792 | Shallow | Sheltered | Lagoon | Park | 12 | 12 | 2 |
| | Turtle Reef | C_TR'15 | 40.0070 | -3.3710 | Deep | Exposed | Fringing | Park | 12 | 12 | 2 |
| | Watamu Coral Garden | C_WCG'15 | 39.9920 | -3.3829 | Shallow | Sheltered | Lagoon | Park | 12 | 12 | 2 |
| South | Kibuyuni | S_KB'15 | 39.3369 | -4.6405 | Shallow | Sheltered | Channel | Unprotected | 24 | 24 | 2 |
| | Makokokwe | S_MK'14 | 39.3361 | -4.7254 | Deep | Exposed | Patch | Park | 16 | 16 | 4 |
| | Upper Mpunguti Leeward | S_UPL'14 | 39.4067 | -4.7026 | Deep | Sheltered | Fringing | Reserve | 12 | 12 | 3 |
| | Upper Mpunguti Seaward | S_UPS'14 | 39.4171 | -4.7002 | Deep | Exposed | Fringing | Reserve | 12 | 12 | 3 |
| | Mkwiro | S_MR'14 | 39.3836 | -4.6575 | Shallow | Sheltered | Channel | Unprotected | 24 | 24 | 3 |
| | Kisite Deep | S_KD'14 | 39.3738 | -4.7142 | Deep | Sheltered | Fringing | Park | 12 | 12 | 3 |
| | Kisite Leeward | S_KL'14 | 39.3666 | -4.7164 | Shallow | Sheltered | Fringing | Park | 12 | 12 | 4 |
| | Kisite Seaward | S_KS'14 | 39.3760 | -4.7228 | Shallow | Exposed | Fringing | Park | 16 | 16 | 4 |

respectively)($F_{2,88} = 9.5$, $p = 0.00$). Density of corals in the 21-40cm size class differed significantly ($F_{2,88} = 15.0$, $p = 0.00$) with highest density in the south (143.3 ± 81.3 colonies) followed

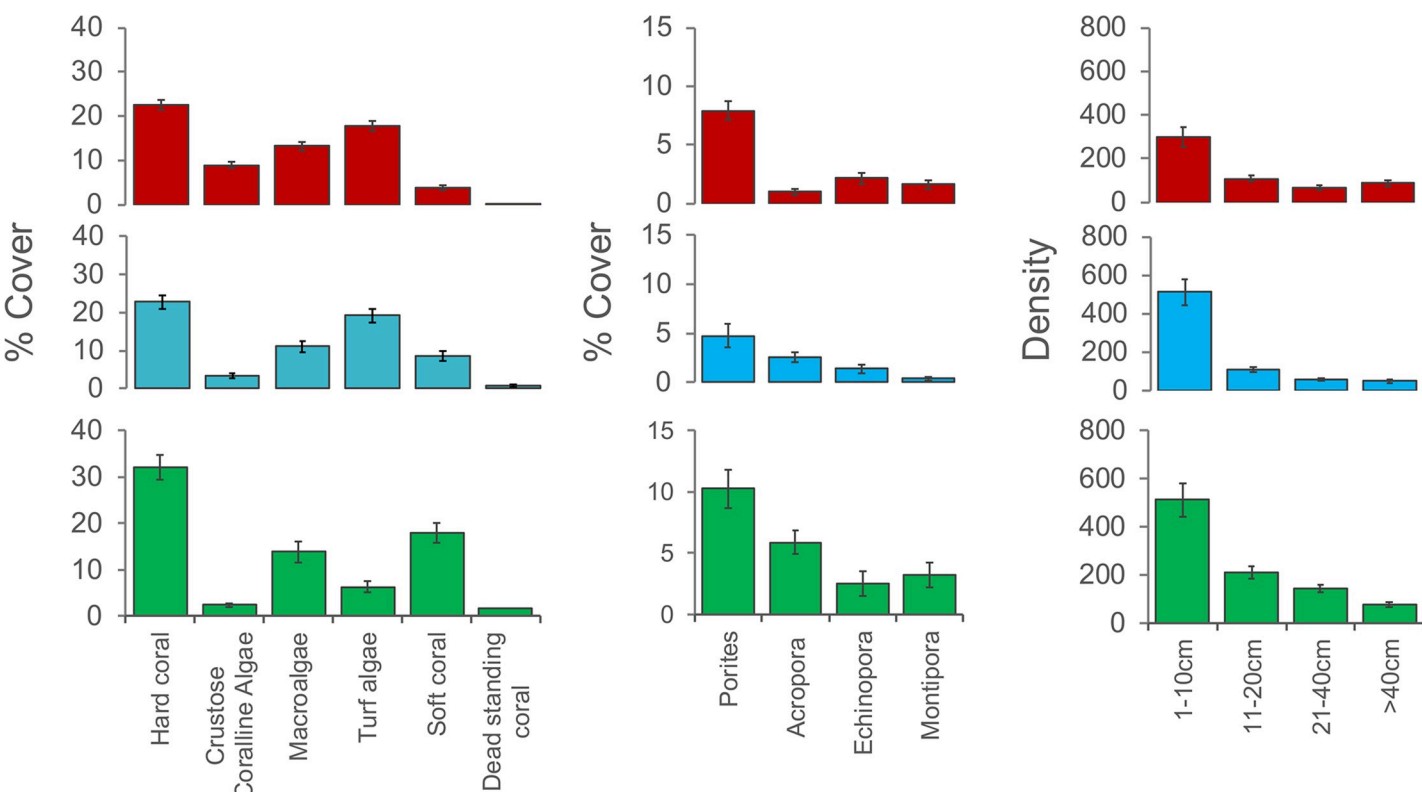

**Fig 3. Summary of benthic community variables by geographic zone.** Mean percentage cover of five major benthic cover categories, four coral genera and density (number of colonies per 100m$^2$) of four coral size classes for north (upper, red), central (middle, blue) and south (lower, green) geographic zones. Error bars indicate standard error.

by north (66.4 ± 65.0 colonies) and central (57.1 ± 30.4 colonies). The size-class >40cm did not show significant difference among geographic zones (Table 2) but densities were higher in the north (86.7 ± 101.0 colonies).

## Habitat factors and management status

A PERMANOVA test on the effect of habitat factors and management showed significant effect ($p < 0.05$) of depth and management on benthic variables (Table 3). While exposure and reef type did not show significant effect, their interaction with depth and management was significant.

Embedded on the four quadrants of a PCoA for the key variables is an illustration of different combinations of habitat and management interactions at the three zones (Fig 4). The upper-left quadrant represents a habitat found only in the north zone with the benthic community highly dominated by crustose coralline algae. This habitat is characterized by Deep-Exposed Patch reefs that occur within Reserves (DEPR). The upper-right quadrant contains sites from central and south zones dominated by soft corals, *Acropora* and *Montipora*, and with a high density of juvenile corals (1-10cm) and small adult corals (11-20cm). This habitat is characterized by Deep-Exposed Fringing reefs within Unprotected areas (DEFU). The lower-left quadrant contains sites found mainly in the north and central zones. The benthic community is predominantly turf algae and macroalgae, and is characterized as Shallow Fringing and Lagoon reefs within Parks and Reserves (SFLPR). The quadrant to the lower-right

**Table 2. ANOVA test on variation of benthic communities across three geographical zones.** Results of One-way ANOVA test done to determine variation of key benthic communities across the 3 geographical zones.

| Benthic community | | Source of Variation | SS | df | MS | F | P-value | F crit |
|---|---|---|---|---|---|---|---|---|
| Major benthic categories | Hard corals | Between Groups | 9681 | 2 | 4840 | 6.884 | 0.001 | 3.007 |
| | | Within Groups | 543512 | 773 | 703 | | | |
| | | Total | 553192 | 775 | | | | |
| | Crustose Coralline Algae | Between Groups | 6583 | 2 | 3291 | 21.022 | 0.000 | 3.007 |
| | | Within Groups | 121023 | 773 | 157 | | | |
| | | Total | 127605 | 775 | | | | |
| | Macroalgae | Between Groups | 669 | 2 | 334 | 0.646 | 0.524 | 3.007 |
| | | Within Groups | 399998 | 773 | 517 | | | |
| | | Total | 400666 | 775 | | | | |
| | Turf algae | Between Groups | 15335 | 2 | 7668 | 16.848 | 0.000 | 3.007 |
| | | Within Groups | 351799 | 773 | 455 | | | |
| | | Total | 367134 | 775 | | | | |
| | Soft corals | Between Groups | 20868 | 2 | 10434 | 44.642 | 0.000 | 3.007 |
| | | Within Groups | 180668 | 773 | 234 | | | |
| | | Total | 201536 | 775 | | | | |
| | Dead standing coral | Between Groups | 207 | 2 | 104 | 15.187 | 0.000 | 3.007 |
| | | Within Groups | 5280 | 773 | 7 | | | |
| | | Total | 5488 | 775 | | | | |
| Coral genera | Porites | Between Groups | 2224 | 2 | 1112 | 3.610 | 0.028 | 3.007 |
| | | Within Groups | 238475 | 774 | 308 | | | |
| | | Total | 240699 | 776 | | | | |
| | Acropora | Between Groups | 2457 | 2 | 1229 | 24.765 | 0.000 | 3.007 |
| | | Within Groups | 38397 | 774 | 50 | | | |
| | | Total | 40854 | 776 | | | | |
| | Echinopora | Between Groups | 99 | 2 | 49 | 0.487 | 0.615 | 3.007 |
| | | Within Groups | 78645 | 774 | 102 | | | |
| | | Total | 78744 | 776 | | | | |
| | Montipora | Between Groups | 644 | 2 | 322 | 5.132 | 0.006 | 3.007 |
| | | Within Groups | 48358 | 771 | 63 | | | |
| | | Total | 49001 | 773 | | | | |
| Coral size classes | 1-10cm | Between Groups | 1005227 | 2 | 502614 | 4.962 | 0.009 | 3.100 |
| | | Within Groups | 8914194 | 88 | 101298 | | | |
| | | Total | 9919422 | 90 | | | | |
| | 11-20cm | Between Groups | 187077 | 2 | 93538 | 9.496 | 0.000 | 3.100 |
| | | Within Groups | 866853 | 88 | 9851 | | | |
| | | Total | 1053930 | 90 | | | | |
| | 21-40cm | Between Groups | 121144 | 2 | 60572 | 14.983 | 0.000 | 3.100 |
| | | Within Groups | 355749 | 88 | 4043 | | | |
| | | Total | 476894 | 90 | | | | |
| | >40cm | Between Groups | 21669 | 2 | 10835 | 1.725 | 0.184 | 3.100 |
| | | Within Groups | 552592 | 88 | 6279 | | | |

contains sites from all geographic zones and has moderate levels of all variables. This represents a habitat that characterized as Shallow Patch and Channel reefs (SPC).

Of the four habitats described from this study, the north and central zones contained three of them and south contained two habitats (S1 Fig). DEPR was found uniquely in the north,

**Table 3. PERMANOVA test on the effect habitat factors and management contributing to the differences in benthic communities.**

| Source of variation | Df | SS | R2 | F | Pr(>F) |
|---|---|---|---|---|---|
| Depth | 1 | 0.137 | 0.090 | 4.934 | 0.003 |
| Exposure | 1 | 0.047 | 0.031 | 1.692 | 0.281 |
| Reef-type | 3 | 0.128 | 0.084 | 1.538 | 0.213 |
| Management | 2 | 0.148 | 0.097 | 2.659 | 0.052 |
| Depth x Exposure | 1 | 0.091 | 0.059 | 3.270 | 0.003 |
| Depth x Reef-type | 1 | 0.042 | 0.028 | 1.525 | 0.185 |
| Exposure x Reef-type | 3 | 0.119 | 0.078 | 1.424 | 0.186 |
| Depth x Management | 2 | 0.088 | 0.057 | 1.581 | 0.094 |
| Exposure x Management | 2 | 0.035 | 0.023 | 0.621 | 0.812 |
| Reef-type x Management | 2 | 0.104 | 0.068 | 1.866 | 0.057 |
| Depth x Exposure x Management | 1 | 0.024 | 0.016 | 0.875 | 0.382 |
| Depth x Reef-type x Management | 1 | 0.094 | 0.062 | 3.395 | 0.032 |
| Residual | 17 | 0.473 | 0.309 | | |
| Total | 37 | 1.530 | 1 | | |

SFLPR was found in all geographic zones, DEFU was found in central and south zones and SPC was found in the north and central zones.

## Discussion

Two distinct patterns of coral reef benthic community structure were observed along the Kenyan coast. A latitudinal gradient was observed indicating a differentiation in benthic community composition between northern and southern sites. A second pattern was based on localised habitat and management factors prevailing within each geographic zone.

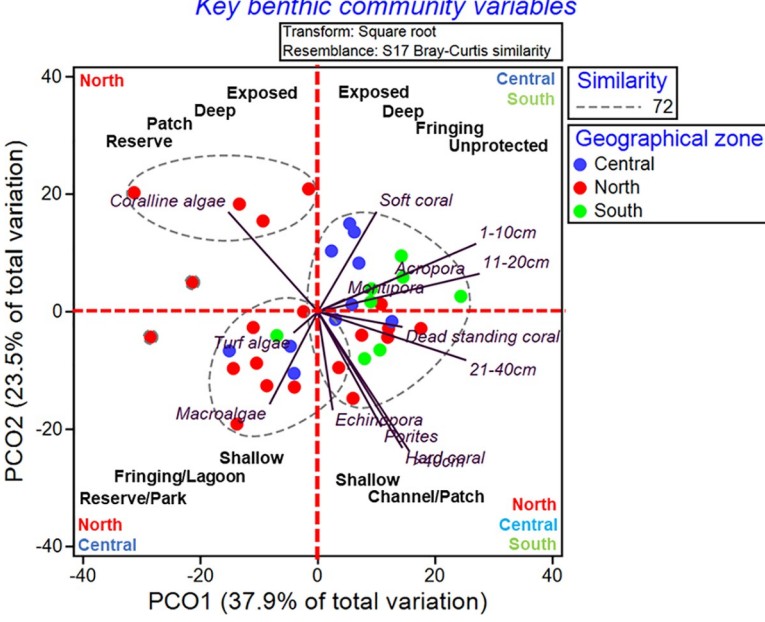

**Fig 4. Principal Coordinate analysis (PCoA) of key benthic communities.** Clusters of 38 reef sites based on Bray–Curtis index. Embedded in each quadrant of the PCO is interpretation of the habitat factors and management levels associated with the clustering of sites.

## Latitudinal/geographic gradients in benthic community composition

The observed differences in benthic community composition among geographic zones found in this study is consistent with previous authors that have documented the biogeographic distribution of reef-building corals [37, 16].

The south zone had a higher LCC with most genera occurring in abundance but mainly dominated by *Porites* and *Acropora*. This area is known to have a high diversity of corals mainly due to its proximity to the WIO center of biodiversity which has been documented to be around the northern Mozambique channel (NMC) [37, 16]. The south zone is the receiving point of coral larvae into Kenyan waters as currents transporting larvae progress northwards [17] resulting in higher coral diversity compared to the north. In the north, a smaller number of genera was found with *Porites* being dominant, indicating a marginal coral community as it is farther from sources of larvae. The north zone is also an area of poor reef development due to the unfavorable conditions caused by the convergence of the colder nutrient rich SC and the EACC, which diminishes chances of high coral diversity in this zone. This convergence of currents makes the north an area of biogeographic transition from southern East African reef communities to Somali-Arabian Gulf communities to the north. This may mean that the north zone can act as an environmental corridor for species movements as the climate warms as has been suggested for other marginal communities [38]. Other studies have reported low coral diversity on marginal reefs e.g. upwelling reefs of the Arabian gulf [39], but there is also some isolated cases of high coral cover and diversity in marginal reefs e.g., at the world's southern-most coral reefs in subtropical Islands of the Pacific Ocean [40, 41]. Some of these marginal reefs have the potential to act as habitable spaces for corals as climate warms, creating climate refugia or 'seed banks' that can re-populate coral communities once climate stabilizes to favorable conditions [42].

The benthic communities in the north and south represent two alternate aspects of coral reef resilience. The comparably lower abundance of juvenile coral colonies in the north and comparably higher abundance of large colonies suggests poor coral recruitment but high survival of adult coral colonies. Low recruitment could be attributed to isolation of the north zone from the southern sources of larvae, with only local sources of larvae for recruitment. Higher abundance of large colonies in the north may be due to the coral community in the north being protected from bleaching mortality by the cooling effect of upwellings in the SC. On the other hand, higher abundance of juvenile corals in the south indicates high recruitment likely due to its proximity to sources of larvae [17, 16]. The low abundance of large colonies in the south implies low survival and only a few coral colonies surviving to large sizes. The south zone is known to face more stress from bleaching as it is closer to the warmest zones at 10–12 ${}^{o}$S, and higher anthropogenic pressures due to higher human population densities as compared to the north. There have been several bleaching episodes since the 1998/99 mass coral bleaching which could have slowed down recovery of coral communities in the south.

Other studies in the WIO region have shown that the coral community in the north is very different from that in the south, with a higher diversity in the south comprising species more closely resembling the northern Tanzanian coral community [43, 16]. Studies from other regions have also shown that the latitudinal gradient drives differences in benthic community composition on coral reefs. Coral reefs in the Hawaiian Archipelago in the north Pacific Ocean show a latitudinal gradient contributing to three distinct reef regimes each dominated by either hard coral, turf algae or macroalgae [44]. In a Caribbean-wide study, spatial and temporal differences in coral cover and macroalgae were found between subregions [45]. Eastern Australia coral reefs showed a trend of decreasing cover of Acroporidae as latitude increased.

The diversity of benthic communities differed along the latitudinal gradient and across geographic zones in Taiwan [46].

## Influence of habitat factors and management on benthic community composition

Habitat factors and management influenced the pattern of benthic communities within each geographic zone. This offers the opportunity to propose ecological controls that structure benthic communities at this scale, and which may be used to guide resource management decisions. Similar findings have been reported in parts of the Pacific Ocean, where benthic communities were structured by depth, exposure to waves and reef types, in the Hawaiian Islands [47–49] and in French Polynesia [50–54].

The heterogeneity and uniqueness of habitat types in the north of Kenya can be attributed to a number of factors including the fact that it is composed of a set of small islands in an archipelago, presenting diverse reef types at different depths. Perpendicular to the shore, in a cross-shelf line, there are numerous channel reefs between the islands and mainland, then lagoons, fringing reefs and at the outermost edge, a deep ridge with patch reefs on it. In addition, the northern part of Kenya has been described as an 'ecotone' zone due to the presence of marginal/transitional coral reef communities composed of rare or regional endemic species [16]. Other studies have also shown significant differences in benthic communities within archipelagos such as distinct coral species assemblages among different islands in Penghu in Taiwan [55], a cross-shelf difference in benthic community within the Spermonde Archipelago in the Coral Triangle [56] and differences in benthic communities across habitats in the Abrolhos archipelago of eastern Brazil [57].

The habitat type DEPR that was uniquely found in the north zone is dominated by coralline algae, which may be due to a high biomass of fish on the patch reefs [21, 58]. Fish herbivores graze away competitors of crustose coralline algae, and fish predators feed on sea urchins which are the greatest eroders of coralline algae [59]. Also, the upwelling currents in this zone could be a source of nutrients stimulating growth of coralline algae. Previous studies have associated coralline algae as a suitable cue for coral settlement and recruitment [60, 61] making these habitats potential spaces for corals to grow. Being deep habitats in an area of upwelling-cooler waters could offer an ecological refugia to coral communities as the climate warms [62]. However, light limitation would reduce the opportunity of a diverse coral community growing and dominating these spaces [63–65] and only a few taxa that can tolerate these depths would colonise this habitat.

A majority of deep- fringing reefs that are exposed to oceanic waves in the central and north zones were dominated by *Acropora*, soft corals and small-sized coral colonies (DEFU habitat type). This habitat type is characterized by gradual slopes on the fringing reefs creating an environment where *Acropora* can dominate competitively. Other studies have reported *Acropora* occupying similar environments of forereef areas, where there is low stress from sedimentation and high rates of development processes such as reproduction [66]. Notably, the low abundance of large colonies within this habitat is worrying considering that the south zone is an area of high coral diversity. This could be indicative of failure to replenish standing stocks of large colonies in the central and south zone, likely due to bleaching mortality which has been recurrent over the past two decades [26, 67]. All zones were badly affected by the 1997/98 mass coral bleaching episode with a 50% - 80% average loss of coral cover with some individual sites loosing up to 100% cover [68, 58]. Recovery has been slow since then perhaps due to the occurrence of smaller-scale bleaching episodes in 2005, 2007, 2010 and 2016 [26, 23, 67]. In the south zone, this bleaching impact was likely compounded by other anthropogenic

stresses originating from its high human settlement. A similar explanation goes for the central zone reef where this study found few large colonies likely due to a slowed down recovery from the mass coral bleaching. Post-bleaching reports indicate that there was an increase in macro-algae and a decrease in fish biomass followed by an explosion in sea urchin density probably as a response to the decrease of fish biomass which constitute their predators. Consequently, sea urchin grazed the reef substate to bare [25]. The presence of poor water conditions due to proximity to river Sabaki effluents, high fishing pressure and recurrent bleaching episodes could have made perseverance of coral colonies to large sizes a great challenge [69, 23].

Shallow fringing or lagoon reefs in the north and central zones (SFLPR habitat type) were characterized by a dominance of macroalgae and turf algae. Most of these habitats are proxi-mal to land and are highly influenced by land-use activities such as effluents from rivers as well as fishing. When hard corals die macroalgae and turf algae take over the space due to increased nutrient availability or reduced herbivory [70–72]. Turf algae are the main focus of grazing but their form allows for rapid regrowth [73] making them thrive even in protected areas where grazing is high. Other studies have found that macroalgae are not influenced by latitudinal effects, with more local-scale influences such as nutrient levels strongly driving their distribu-tion [74]. In this study, the abundance of macroalgae within these habitats indicates a worrying level of coral reef degradation since most coral cover and taxonomic diversity has always been thought to occur within shallow depths [75–77].

Shallow habitats on patch or channel reefs (SPC) were dominated by large colonies with an abundance of *Porites* and *Echinopora*. Similar observations were made in the Great Barrier Reef where large colonies of *Porites* were associated with shallow inshore reefs [78]. The pres-ence of large coral colonies in shallow reefs indicates a habitat type that supports the survival of corals probably due to acclimatization to bleaching [79]. Also, the high fluctuation of water temperatures in shallow reefs creates a variable environment that may support different coral genotypes. A diversity of genotypes offers a chance for coral colonies to survive to larger sizes and persist over time as seen in an inshore reef of the Great Barrier Reef where *Porites* domi-nated through very persistent genotypes [80]. In addition, *Porites* and *Echinopora* have been considered as resistant and generalist coral genera within the functional groups of corals, based on their response to bleaching [81]. This makes them capable of inhabiting shallow areas where competitive but less tolerant corals cannot survive the fluctuating temperatures, bleaching events and proximity to human disturbance such as sedimentation.

*Porites* is a widespread genus that occurs at different habitats and thus reinforces its func-tional role as a resistant species [81–83]. The presence of very large colonies of *Porites* in the north compared to the south could imply that bleaching mortality of *Porites* is lower in the north due to the presence of habitat types that support resistance properties for that genus. In the south zone, the high percentage cover of *Porites* is mainly composed of small colonies. This signifies an area which could be facing high bleaching mortality, as evidenced by high percent-age cover of dead standing corals, but experiences high recruitment that recovers the standing stock. This recovery feature may not be present in the north because *Acropora* communities failed to recover after the 1998 coral bleaching episode and recruitment is persistently low [21]).

## Implications for conservation

In order to effectively manage and mitigate the degradation of coral reefs, information on community spatial patterns, habitat and management influences is necessary. Bleaching epi-sodes have become more frequent and under the most recent IPCC report, 99% of coral reefs are expected to be lost this century if temperatures increase 2°C above the pre-industrial

baseline [6]. However, if temperatures can be stabilized at 1.5˚C, then 10~30% of the reefs could be saved [84]. These reefs could serve as refugia [62, 85] or 'seed banks' to repopulate other reefs once mechanisms of reducing climate change to desirable levels have been achieved [86].

Management strategies should ensure the protection of habitats that exhibit bleaching protection or recovery characteristics in order to enhance the resilience of coral communities. In addition, there should be mechanisms to reduce local impacts such as sedimentation and pollution. High sedimentation levels are experienced on near-shore coral reefs near rivers with high sediment load, particularly in the central zone due to the river Sabaki and river Tana [87]. Most of the reefs in this area are within parks and reserves but the effect of sediment load is likely to still degrade these reefs. In the south zone, sediment loading is mainly experienced from smaller rivers and creeks, with increasing threat from logging of mangroves, thus exposing sediments to erosion. In all these areas, increasing urbanization elevates discharge of sewage and other pollutants, further exacerbating stress to corals.

In conclusion, the occurrence of a geographical and a local habitat pattern in coral benthic communities that is further mediated by the level of management offers a range of environments that support coral reef communities. The north zone presents a unique environment of a marginal reef that provides protection to coral communities from bleaching [88] even as the climate warms, expressing resistance properties of coral reef resilience [89]. The south zone's high diversity of corals [16] and high recruitment potentially offer a habitat that expresses recovery properties after a bleaching disturbance [88]. The central zone is intermediate, providing a corridor for transfer of coral propagules between the north and south. Similar propositions on the importance of ecological corridors in conservation have been mentioned for large scale studies in the Eastern Tropical Pacific [90] and within the Gulf of Mexico [91].

The heterogeneity of coral reef habitat types in Kenya offers an opportunity for designing MPAs that protect a diverse range of functional traits of coral communities [92]. As such, spatial heterogeneity in MPAs may reduce the risk of catastrophic regime shifts [93].

## Supporting information

**S1 Table. Sampling design.** A table of sites sampled according to their geographical zones, habitat factors and management levels.
(DOCX)

**S2 Table. Benthic cover data.** Percentage cover of major benthic categories recorded at all surveyed coral reef sites along the Kenyan coast.
(DOCX)

**S3 Table. Coral genera data.** Percentage cover of coral genera recorded at all surveyed coral reef sites along the Kenyan coast.
(DOCX)

**S4 Table. Coral colony size class data.** Density (no. of colonies per 100m$^2$) of coral colonies recorded at all surveyed coral reef sites along the Kenyan coast.
(DOCX)

**S5 Table. Mean percentage cover and standard deviation of 15 major benthic categories.** Summary of all studied sites along the Kenyan coast.
(DOCX)

**S6 Table. Mean percentage cover and standard deviation of 44 coral genera.** Summary of all studied sites along the Kenyan coast.
(DOCX)

**S7 Table. Mean density of coral size classes (number of colonies per 100m$^2$).** Summary of all studied sites along the Kenyan coast.
(DOCX)

**S8 Table. PERMANOVA main-test table of results.**
(DOCX)

**S1 Fig. Illustration of the four habitat types occurrence at the three geographical zones.** North zone has three habitat types one of them uniquely to this zone (DEPR); Central zone has three habitat types, one it shares with all other zones (SPC), another with only north (SFLPR) and only south zone (DEFU); South zone has only two habitat types, one it shares with all zones (SCP) and the other with only Central zone (DEFU).
(TIF)

## Acknowledgments

We would like to particularly thank Benjamin Cowburn for his assistance in data collection, James Mbugua for assisting the development of the study site map, Maxwel Azalli for advising on data analysis and Gladys Okemwa for helpful discussions during the development of this study.

## Author Contributions

**Conceptualization:** Juliet Furaha Karisa, David Obare Obura.

**Formal analysis:** Juliet Furaha Karisa.

**Funding acquisition:** Juliet Furaha Karisa, Chaolun Allen Chen.

**Methodology:** Juliet Furaha Karisa, David Obare Obura.

**Supervision:** David Obare Obura, Chaolun Allen Chen.

**Writing – original draft:** Juliet Furaha Karisa.

**Writing – review & editing:** David Obare Obura, Chaolun Allen Chen.

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
