## [Decision Letter · Decision Letter 0]

25 Jun 2020

PONE-D-20-06669

Spatial heterogeneity of coral reef benthic communities in Kenya

PLOS ONE

Dear Dr. Karisa,

Thank you for submitting your manuscript to PLOS ONE. After careful consideration, we feel that it has merit but does not fully meet PLOS ONE’s publication criteria as it currently stands. Therefore, we invite you to submit a revised version of the manuscript that addresses the points raised during the review process. Please submit your revised manuscript by Aug 09 2020 11:59PM. If you will need more time than this to complete your revisions, please reply to this message or contact the journal office at plosone@plos.org. Please include the following items when submitting your revised manuscript:

We look forward to receiving your revised manuscript.

Kind regards,

Jose M. Riascos, Ph.D.

Academic Editor

PLOS ONE

Journal Requirements:

2. In your Methods section, please provide additional location information of the study sites, including geographic coordinates for the data set if available.

3. In your Methods section, please provide additional information regarding the permits you obtained for the work. Please ensure you have included the full name of the authority that approved the study site access and, if no permits were required, a brief statement explaining why.

Reviewers' comments:

Reviewer's Responses to Questions

**Comments to the Author**

1. Is the manuscript technically sound, and do the data support the conclusions?

Reviewer #1: Yes

Reviewer #2: Yes

2. Has the statistical analysis been performed appropriately and rigorously? 

Reviewer #1: Yes

Reviewer #2: Yes

3. Have the authors made all data underlying the findings in their manuscript fully available?

Reviewer #1: Yes

Reviewer #2: Yes

4. Is the manuscript presented in an intelligible fashion and written in standard English?

Reviewer #1: Yes

Reviewer #2: Yes

5. Review Comments to the Author

Reviewer #1: I have read and reviewed the manuscript by Karisa et al. In this study the authors use multivariate analysis of extensive data from 38 reef sites of different geographical zones, reef habitats and management level to investigate the effects of these factors on structuring the spatial patterns in benthic communities of the extensive Kenyan coast.

They found a two distinct patterns of coral reef benthic community structure along the coast: A latitudinal gradient between northern and southern sites, and a second pattern based on the influence of habitat and management factors within each geographical zone.

The authors identified four habitats in the three geographical zones based on their benthic community composition similarity, one of the habitats was found across the three zones, one in the north and central zones, one in the central and south zones and one exclusively in the north zone. Also, they concluded that the north zone habitats support resistance properties because is a zone exposed to more extreme conditions, the south zone supports recovery properties and the central zone acts as an ecological corridor between zones.

I found the manuscript interesting, generally well written, however, I have some concerns which I describe in detail below.

General comments:

-Study area. I suggest to the authors that include the sea temperature ranges in the northern and southern areas, even though the temperature is not a variable in the analyzes, it is important in the discussion, since it seems that in the reefs of the southern zone the temperature ranges are narrower and more stable than in the north, which makes them more susceptible to bleaching during episodes of high temperature than the north.

Specific Comments:

-Line 18 (page 2): the number of reefs sampled was 38 not 39, change.

-Line 111 (page 6): in the S1Table there is an error in the number of reefs in the reef types, they sum 21, and in the zone total you considered only 20, verify please.

-Line 138, 139 (page 7): check the supplementary tables numbers, S1 corresponds to S2; S2 to S3 and S3 to S4.

Line 171 (page 9): S2 Table corresponds to S5 Table

Line 174 (page 9): S3 Table corresponds to S6 Table

Line 177 (page 9): S4 Table corresponds to S7 Table

Line 193 (page 9) and line 195 (page 10). Fig 2B, the order of the graphs in the figure are wrong, 2B is size-classes and 2C is coral genera, I suggest making the correction in the figure not in the text.

Line 194 (page 10): is S6 Table not S3 Table

Line 199 (page 10): is 38 not 39 reefs sites

Line 203 (page 10): is S8 Table not S5 Table

Line 277 (page 16): is 38 not 39 reef sites

Line 281 (page 16): is S9 Fig not S6 Fig.

Line 357 (page 19): I think a missing word here “attributed to a number of factors including the that it is composed...”

Reviewer #2: This is a good paper which could be published following very little modification. It does appear to focus, in the way it is written, on the analysis while the point of doing it, which is to determine the spatial heterogeneity and coral benthic communities as noted in the title is dealt with a little bit too briefly. Also the management implications of this I think could be expanded a little. But it seems to be sound, the data are well analysed, and the conclusions are justified. I wonder if another graphic on a map with different zones might be possible to help us visualise the picture a little better.

Some specifics:

Lines 180-181. Not sure what is meant here. Isn’t Halimeda a valid inclusion?

Lines 191. You need a sentence to better explain why only these four coral general were used.

In general here, given the increasing frequency of ocean heat waves, I would think if you can increase the importance and analysis of juvenile corals that might be sensible. Often juveniles, an indication of recent fecundity and coral health, are regarded as those less than 1 cm across which you largely ignore I think. You might explain why you left these out even if it is simply to say that it was impractical given your time.

Lines in Discussion 348-440. I felt this was a little wordy and could be cut down where it is just re-describing things already said in the results. Not essential but I think it would improve things a bit. Conversely, I thought the next bit of the discussion on management implications could have benefited with a bit more discussion!

6. PLOS authors have the option to publish the peer review history of their article (what does this mean?). If published, this will include your full peer review and any attached files.

Reviewer #1: No

Reviewer #2: **Yes: **Charles Sheppard

---

## [Author Response · Author response to Decision Letter 0]

23 Jul 2020

A. JOURNAL REQUIREMENTS

→We edited the manuscript and followed PLOS ONE style requirement. We also named the files submitted according to the format required by the journal.

In your Methods section, please provide additional location information of the study sites, including geographic coordinates for the data set if available

→We added geographic coordinates for all studied sites in Table 1 by providing the latitudes and longitudes for each site. 

In your Methods section, please provide additional information regarding the permits you obtained for the work. Please ensure you have included the full name of the authority that approved the study site access and, if no permits were required, a brief statement explaining why

→This study did not require a research permit because of existing memorandum of understanding between the conservation institutions and the lead author’s institution (which is a national institution) and therefore allows for research without a permit.

B. REVIEWERS’ COMMENTS

Reviewer #1

General comments:

Study area. I suggest to the authors that include the sea temperature ranges in the northern and southern areas, even though the temperature is not a variable in the analyzes, it is important in the discussion, since it seems that in the reefs of the southern zone the temperature ranges are narrower and more stable than in the north, which makes them more susceptible to bleaching during episodes of high temperature than the north.

→We included information on Sea Surface Temperature (SST) of the three geographic zones. Mean, standard deviation, maximum and minimum SST for each zone was derived from MODIS satellite by averaging SST for two ‘normal’ years (2013-2014) which had minimal interference from extreme events e.g the Indian Ocean Dipole. We considered pixels that were quite far off the shore to avoid SST variations influenced by the land mass.

Specific comments:

Line 18 (page 2): the number of reefs sampled was 38 not 39, change.

→There was an error, the number of reefs sampled was 38 and not 39. We corrected this in the text. 

Line 111 (page 6): in the S1Table there is an error in the number of reefs in the reef types, they sum 21, and in the zone total you considered only 20, verify please.

→There was an error, the number of fringing reefs in north zone was 5 not six. We corrected this in S1 Table.

Line 138, 139 (page 7): check the supplementary tables numbers, S1 corresponds to S2; S2 to S3 and S3 to S4.

→There was an error in the numbering of supplementary table files for S2 and S4. We have corrected this by changing file names S2 Table = S4 Table, and S4 Table = S2 Table.

Line 171 (page 9): S2 Table corresponds to S5 Table

→These are actually supplementary tables with different information. S2 Table shows the site averages (percentage cover) for only 10 selected major benthic categories. S5 Table shows overall averages (percentage cover) and standard deviations of all major benthic categories recorded in this study. We have renamed the S5 table by adding the word ‘overall’ to differentiate the content in the two tables.

Line 174 (page 9): S3 Table corresponds to S6 Table

→We have renamed the S6 Table by adding the word ‘overall’ to differentiate the content in the two tables

Line 177 (page 9): S4 Table corresponds to S7 Table

→We have renamed the S7 Table by adding the word ‘overall’ to differentiate the content in the two tables

Line 193 (page 9) and line 195 (page 10). Fig 2B, the order of the graphs in the figure are wrong, 2B is size-classes and 2C is coral genera, I suggest making the correction in the figure not in the text.

→We have corrected the graph arrangement in Fig 2.

Line 194 (page 10): is S6 Table not S3 Table

→We corrected the error 

 

Line 199 (page 10): is 38 not 39 reefs sites

→We corrected the error

Line 203 (page 10): is S8 Table not S5 Table

→We corrected the error

Line 277 (page 16): is 38 not 39 reef sites

→We corrected the error

Line 281 (page 16): is S9 Fig not S6 Fig.

→We corrected the error

Line 357 (page 19): I think a missing word here “attributed to a number of factors including the that it is composed...”

→We added the missing word ‘fact’

Reviewer #2

This is a good paper which could be published following very little modification. It does appear to focus, in the way it is written, on the analysis while the point of doing it, which is to determine the spatial heterogeneity and coral benthic communities as noted in the title is dealt with a little bit too briefly. Also, the management implications of this I think could be expanded a little. But it seems to be sound, the data are well analysed, and the conclusions are justified. I wonder if another graphic on a map with different zones might be possible to help us visualise the picture a little better.

→We included a section in the discussion to indicate the implication of spatial patterns to management of coral reefs in Kenya. Here, we expanded the implication of these spatial patterns Based on the current management efforts on coral reefs in Kenya. We also added information about the study sites by describing the different levels of coral reef management at the three geographic zones. However, we could not add a map with additional graphic representation because the national scale of the Kenyan map makes it impossible to show the zonation on a 100m scale. The reef zones are also very closely associated with one another, though with some differences as found in the paper. Instead, we added more information by describing the study sites based on their geomorphological zones.

Specific comments:

Lines 180-181. Not sure what is meant here. Isn’t Halimeda a valid inclusion?

→Based on Fig 2A, the direction of vectors on the PCoA show that crustose coralline algae (CCA) and Halimeda pointed to the same direction. This indicated they influenced the clustering of similar sites. In order to select just a few representative variables to determine the spatial patterns, we selected only CCA based on this rationale. We have revised the explanation in the text to clearly describe this process.

Lines 191. You need a sentence to better explain why only these four coral genera were used.

→We provided more explanation on the criteria we used to select key variables for the coral genera.

In general, here, given the increasing frequency of ocean heat waves, I would think if you can increase the importance and analysis of juvenile corals that might be sensible. Often juveniles, an indication of recent fecundity and coral health, are regarded as those less than 1 cm across which you largely ignore, I think. You might explain why you left these out even if it is simply to say that it was impractical given your time.

→It would have been impossible to get reliable data for colonies below 1cm using the methodology in this study. Quadrats that can measure less than 1cm colonies were not included in the method as described in Obura and Grimsditch (2009) and citations within. The method in this study assumes that colonies less than 10cm in diameter are not reproductive, so colonies are described as ‘juveniles’ and, in most cases, not reproductive. Adequate information on reproductive maturity for most coral species is lacking, particularly for this region, so we acknowledge this is an approximation.

Lines in Discussion 348-440. I felt this was a little wordy and could be cut down where it is just re-describing things already said in the results. Not essential but I think it would improve things a bit. Conversely, I thought the next bit of the discussion on management implications could have benefited with a bit more discussion!

→In this section, we discussed the four habitats that have been identified in this study. This discussion was important as we tried to use the information to derive ecological implication of having these four habitats. This information was also critical in deriving the discussion on management implication. In order to address this concern, we added more information on the management implication section. We focused on Kenya-specific management implication based on the spatial pattern found in this study.

---

## [Editor Report · Decision Letter 1]

27 Jul 2020

Spatial heterogeneity of coral reef benthic communities in Kenya

PONE-D-20-06669R1

Dear Dr. Karisa,

We’re pleased to inform you that your manuscript has been judged scientifically suitable for publication and will be formally accepted for publication once it meets all outstanding technical requirements.

Kind regards,

Jose M. Riascos, Ph.D.

Academic Editor

PLOS ONE
---

## [Editor Report · Acceptance letter]

30 Jul 2020

PONE-D-20-06669R1 

Spatial heterogeneity of coral reef benthic communities in Kenya 

Dear Dr. Karisa:

I'm pleased to inform you that your manuscript has been deemed suitable for publication in PLOS ONE. Congratulations! Your manuscript is now with our production department. 

Kind regards, 

on behalf of

Professor Jose M. Riascos 

Academic Editor

PLOS ONE